# A Novel Approach to 1RM Prediction Using the Load-Velocity Profile: A Comparison of Models

**DOI:** 10.3390/sports9070088

**Published:** 2021-06-22

**Authors:** Steve W. Thompson, David Rogerson, Alan Ruddock, Leon Greig, Harry F. Dorrell, Andrew Barnes

**Affiliations:** 1Academy of Sport and Physical Activity, Sheffield Hallam University, Sheffield S10 2BP, UK; d.rogerson@shu.ac.uk (D.R.); a.ruddock@shu.ac.uk (A.R.); a.barnes@shu.ac.uk (A.B.); 2School of Health Sciences, Robert Gordon University, Aberdeen AB10 7QE, UK; l.greig12@rgu.ac.uk; 3School of Sport and Exercise Sciences, University of Lincoln, Lincoln LN6 7TS, UK; hdorrell@lincoln.ac.uk

**Keywords:** load-velocity profiling, 1RM prediction, 1RM estimation, maximal strength, linear regression

## Abstract

The study aim was to compare different predictive models in one repetition maximum (1RM) estimation from load-velocity profile (LVP) data. Fourteen strength-trained men underwent initial 1RMs in the free-weight back squat, followed by two LVPs, over three sessions. Profiles were constructed via a combined method (jump squat (0 load, 30–60% 1RM) + back squat (70–100% 1RM)) or back squat only (0 load, 30–100% 1RM) in 10% increments. Quadratic and linear regression modeling was applied to the data to estimate 80% 1RM (kg) using 80% 1RM mean velocity identified in LVP one as the reference point, with load (kg), then extrapolated to predict 1RM. The 1RM prediction was based on LVP two data and analyzed via analysis of variance, effect size (*g*/$\eta_{p}^{2}$), Pearson correlation coefficients (*r*), paired *t*-tests, standard error of the estimate (SEE), and limits of agreement (LOA). *p* < 0.05. All models reported systematic bias < 10 kg, *r* > 0.97, and SEE < 5 kg, however, all linear models were significantly different from measured 1RM (*p* = 0.015 <0.001). Significant differences were observed between quadratic and linear models for combined (*p* < 0.001; $\eta_{p}^{2}$ = 0.90) and back squat (*p* = 0.004, $\eta_{p}^{2}$ = 0.35) methods. Significant differences were observed between exercises when applying linear modeling (*p* < 0.001, $\eta_{p}^{2}$ = 0.67–0.80), but not quadratic (*p* = 0.632–0.929, $\eta_{p}^{2}\;$= 0.001–0.18). Quadratic modeling employing the combined method rendered the greatest predictive validity. Practitioners should therefore utilize this method when looking to predict daily 1RMs as a means of load autoregulation.

## 1. Introduction

A one repetition maximum (1RM) is defined as the maximum external load (kg) an individual can lift for a single repetition [1]. Further, 1RM tests have excellent reliability, relationships with biomechanically similar sporting movements (e.g., back squat and jumping), and can serve as an effective prescriptive tool (% 1RM) [1,2,3,4]. Despite this, large demand is placed on the neuromuscular system, often rendering regular 1RM testing infeasible, particularly in multi-faceted sports (e.g., team or court) due to the importance of technical training, busy competitive schedules, and travel [5]. Frequent maximum testing could therefore create unwanted fatigue, potentially impacting on performances throughout the year [5]. While this is unlikely to be problematic in settings where 1RMs are relatively stable (e.g., strength sports), maximum strength might fluctuate in athletes competing in these sports due to training priorities [5], sleep [6], nutrition [7], and/or fatigue [8]. As a result, alternative strategies such as 1RM prediction from load-velocity profile (LVP) data might be an effective strategy to manipulate load (i.e., autoregulation), which is thought to be vital to optimize athletic development [9].

Construction of a LVP is based on a near perfect relationship (*r* > 0.9) between load (kg or % 1RM) and velocity (mean, peak, or mean propulsive) which facilitates the development of a statistical model (e.g., linear regression) designed to predict load and/or velocities through extrapolation [10,11]. There is an extensive body of literature investigating the validity and reliability of LVPs to predict 1RM across key exercises such as the bench press [12,13,14], back squat [10], deadlift [15,16] prone bench-pull [17], half squat [18,19,20], and leg press [19]. Formative work by Gonzalez-Badillo et al. [21] concluded, amongst others since, that generalized predictive equations were effective in estimating relative load, reducing the need to repeatedly assess maximal strength. More recent research, however, has demonstrated large between-participant variability in velocity [11,22], limiting the application of these generalized models and suggesting individualized LVPs might provide better estimations of submaximal and maximal load.

The multiple-point method [14,17,23,24], where models are built using velocity data from multiple incremental, submaximal loads (e.g., 45–85% 1RM in 10% increments) is a common technique to predict 1RM. Similarly, a simplified two-point version has also been suggested, where 1RM is predicted from two submaximal loads (e.g., 45% and 85% 1RM) [17,25]. Despite differences in the construction of each approach, practically perfect correlations (*r* > 0.9), goodness of model fit (R^2^ > 0.9), and low systematic bias between direct and predicted 1RM data (<10 kg) have been observed [12,14,17,23,24,26]. Whilst these data indicate predictive validity, the studies are limited to isolated, controlled upper body exercises such as the bench press or prone row, rendering the applicability to exercises beyond these unclear.

The predictive validity of the aforementioned modeling approaches in lower body exercises, such as the back squat [19,27], half-squat [18,19,20], and leg press [19], are more equivocal. Coefficient of variations (CVs) of up to 12% between predicted and actual 1RMs have been observed, and a wider range of model fit to observed data (R^2^ = 0.79–0.99) reported, indicating possible model accuracy issues for larger, complex movements. In addition to the heterogeneity in results, all the above research (except Garcia-Ramos et al. [17]) have utilized Smith-machine exercises, limiting the practical recommendations to more applied settings that utilize free-weight exercises.

Despite variety in the construction of the profile (two-point vs. multi-point; start and end loads etc.), most 1RM prediction studies have one similarity: the velocity recorded at 1RM (V_1RM_) as the endpoint of extrapolation. Typically, this value is established either through a direct measure of the V_1RM_ as part of a full profile, or taken from normative data of a similar population, both of which have fundamental flaws for 1RM prediction. The LVP is highly individualized [11,22] and the use of normative velocity data as the endpoint of extrapolation demonstrates large systematic error. Additionally, poor within-participant reliability of V_1RM_ has resulted in large random error in modeled estimates being observed [11,16,22]. While the V_1RM_ appears to be unreliable, previous research has shown the velocity observed at submaximal loads demonstrates better reliability [11]. A combination of increased movement variability, small horizontal movements, and a larger contribution of the stretch-shortening cycle could explain this poorer reliability at V_1RM_ [11,22]. Therefore, an alternative approach might be to incorporate a more stable velocity value (e.g., 80% 1RM) as the method of extrapolation for predicting 1RM, potentially reducing the magnitude of error between modeled and directly assessed 1RM values.

When considering sources of prediction error, the statistical model used to generate the LVP must also be evaluated, with linear regression typically used to create the model. To date, only one study has compared linear regression with an alternative approach (polynomial modeling) for estimating 1RM to determine whether the additional flexibility afforded by this method improves predictive validity [28]. Janicijevic et al. [28] observed a better predictive validity for the multiple-point linear model when compared to polynomial modeling using the Smith-machine bench press exercise. However, despite relatively small mean differences (2.5–4.1 kg), strong correlation coefficients (*r* > 0.95), small effect sizes (<0.2), and small mean systematic bias (−3.2 to −1.0 kg), large random error was observed in all models (20 kg in some cases). Such large random error raises concerns over the utility of these 1RM predictive models as the repeatability and potential to control for noise might be compromised and thus, further comparisons are required.

Typically, LVPs are constructed using a combination of light and heavy loads (30% 1RM to 100% 1RM) in a non-ballistic exercise [11,29]. Despite this, ballistic equivalents (loaded jump squat) are often more commonly prescribed than non-ballistic exercises at these lighter loads (e.g., bodyweight to 60% 1RM) given the greater mechanical outputs, closer relationship with specific sporting actions (i.e., jumping), and larger periods of positive acceleration [30,31,32,33,34]. Therefore, by utilizing both ballistic and non-ballistic exercises within LVPs, arguably a more reliable, valid, and practically representative model could be developed, enabling greater usability in practice. Furthermore, coupling this more valid data with the sophistication of quadratic modeling might offer improved predictions for a complex, free-weight movement such as the back squat. Therefore, the aim of this study was to investigate whether 1RM could be predicted from load-velocity data. Specifically, to compare whether exercise selection (back squat vs. jump squat and back squat ‘*combined*’ method) and model construction (linear vs. quadratic) effects the predictive validity of the LVP using a novel method of extrapolation (80% 1RM) to estimate maximum strength.

## 2. Materials and Methods

### 2.1. Subjects

Fourteen healthy, strength-trained (relative strength > 1.5 × body mass) men (age: 26.0 ± 3.8 years; body mass: 82.5 ± 9.4 kg; stature: 174.7 ± 4.6 cm; relative strength: 1.95 ± 0.2 kg·bm^−1^) volunteered for this study. Ethical approval was granted via the institution ethics board (ER13605026) in accordance with the seventh revision (2013) of the declaration of Helsinki. In addition to relative strength, 12 months resistance training experience and technical proficiency in the free-weight back squat and loaded jump squat exercises were required. Written and verbal informed consent was provided prior to testing.

### 2.2. Procedures

Subjects attended the laboratory on three occasions, each separated by a minimum of 72 h. No additional lower body exercise was permitted 48 h prior to and during data collection. All repetitions were performed using an International Weightlifting Federation approved, calibrated 20 kg barbell and competition bumper plates (Werksan, Akyurt, Turkey). A high-bar back squat technique was adopted which involved the barbell sitting on the upper part of the trapezius muscles and using a neutral grip. Subjects self-selected hip width and foot position, which was recorded and standardized across sessions. A lift was deemed successful when the hip was below the knee at minimum displacement and the lower limbs were fully extended upon ascent. The jump squat was standardized identically to the back squat, but subjects were required to fully leave the floor following ascent. Technique and depth were assessed by an experienced, accredited Strength and Conditioning (S&C) coach and retrospective 2D video analysis (iPhone 7, iOS 14.4.4, Apple, Cupertino, CA, USA) to ensure repetition depth was consistent. The dip function in the Gymaware Linear Position Transducer (Version 2.9.4, Kinetic, Canberra, Australia) was also used to check displacement of the barbell.

### 2.3. 1RM Testing (Visit 1)

Body mass (kg) (Kistler, 9286A, Winterthur, Switzerland), stature (cm) (Seca, Leicester, Hamburg, Germany), and current 1RM estimation were collected during the initial visit. An individualized, standardized warm-up was then performed using a combination of static stretching, dynamic mobility, activation exercises, light barbell work, and body-weight jumps. Habituation of performing the concentric phases with ‘*maximal intent and velocity*’ also occurred.

Subjects were then taken through an incremental 1RM protocol in the free-weight back squat consisting of performing repetitions across a series of incremental loads: 50% (five repetitions); 70% (three repetitions); 80% (two repetitions); 85%, 90%, and 95% (one repetition) of the estimated 1RM followed by up to five attempts to find a true 1RM. 1RM was determined when the subject and primary researcher agreed no more weight could be lifted, or a failed attempt occurred. In addition, 3–5 min rest was prescribed in between each load.

### 2.4. Load-Velocity Profile (Visits 2 and 3)

Visits two and three were procedurally identical. Subjects performed an incremental LVP in the back squat and jump squat exercises. All loads were determined as a percentage of the back squat 1RM from visit one. Gymaware (sampling every 2 mm of displacement) and a 4th generation iPad mini (iOS 14.0.1, Apple, Cupertino, CA, USA) were used to measure mean velocity for each repetition [29]. The Gymaware was located on the right collar, 10 mm from the end of, and perpendicular to, the barbell.

Prior to data collection, subjects completed the same standardized warm-up from visit one in addition to bodyweight repetitions (using a wooden dowel) in the back squat and jump squat. The following loads were then performed sequentially in both exercises: 0 load (five repetitions), followed by 30%, 40% (three repetitions), 50% and 60% 1RM (two repetitions). The participants then continued with back squat only for loads 70% (two repetitions), 80%, 90% and 100% 1RM (one repetition). Participants were given up to three attempts to lift the 1RM achieved in visit one. Five minutes rest was administered between loads, with three minutes between exercises at each load. Subjects were instructed to perform the concentric phase of every repetition with ‘*maximal intent and velocity*’. Mean velocity was defined as the average velocity recorded across the full concentric phase of both exercises. The start and end point of the concentric phase was defined as per the manufacturer’s data processing and filtering system.

### 2.5. 1RM Prediction

The models and methods employed in the present study have five novel factors: (1) the utilization of 80% 1RM mean velocity as the constant (reference point) within the predictive equations; (2) a comparison between linear and quadratic predictive models; (3) a combination of ballistic (jump squat) and non-ballistic (back squat) free-weight exercises compared to non-ballistic (back squat) only; (4) a combination of interpolation and extrapolation to estimate maximal load; and (5) model validation by using one set of data to fit the model and then a new set of LVP data to predict 1RM.

Eight LVPs were created for each individual following data collection (Table 1). The combined method mean velocity data were utilized for four of the profiles, with back squat only mean velocity data applied for the other four. Moreover, a four-point (e.g., combined (quadratic 4)) and seven-point (e.g., back squat (linear 7)) profile was produced for each of the conditions (Table 1). Velocity data for loads between 0 load and 60% 1RM were taken from the jump squat, with anything heavier taken from the back squat when constructing the combined models. All velocity data (0 load to 100% 1RM) were taken from the back squat when constructing the back squat models. A quadratic or linear function was then applied to the data. Models were fit using absolute load (kg) as the independent variable, and mean velocity (m·s^−1^) as the dependent variable. The LINEST function was used in Microsoft Excel (Microsoft Excel, Microsoft, Albuquerque, NM, USA) to determine model parameters for both the quadratic and linear functions. Both equations were then rearranged to solve x:(1)$$\mathrm{Quadratic}\;\mathrm{model}:\;y=ax^{2}+bx+c\;\to \;x=\frac{-b\pm \sqrt{b^{2}-4ac}}{2a}$$
(2)$$\mathrm{Linear}\;\mathrm{Model}:\;y=ax+b\;\to \;x=\frac{y-b}{a}$$

The mean velocity at 80% 1RM was taken from session one and applied to session two’s profiling data, acting as the reference velocity for each model—i.e., estimating kg’s that corresponded to 80% 1RM mean velocity—via a method of interpolation. Further, 80% 1RM was selected as the reference velocity as previous literature has found this to be the heaviest load demonstrating acceptable reliability of mean velocity [11]. Then, 1RM was predicted via a method of extrapolation from 80% to 100% 1RM using absolute (kg) and relative (% 1RM) load only. This was achieved by simply increasing the predicted absolute load (80% 1RM equivalent) by 20% to equate to the predicted 1RM load. Examples of the predictive models can be seen in Figure 1.

### 2.6. Statistical Analysis

All data were assessed for normal distribution and relevant model assumptions for linear and quadratic variants. The predictive validity of each model was assessed by comparing estimated values to measured 1RMs using paired samples *t*-tests, Hedges *g* effect sizes (ES), limits of agreement (LOA), Pearson *r* correlation, and standard error of the estimate (SEE). ES magnitudes were interpreted as: trivial (<0.2); small (0.2–0.59); moderate (0.6–1.19); large (1.19–2.0); very large (>2.0) [35]. Pearson *r* magnitudes were interpreted as: trivial (<0.1); small (0.1–0.29); moderate (0.3–0.49); high (0.5–0.69); very high (0.7–0.89); and practically perfect (>0.9) [11]. A two-way repeated measures analysis of variance (ANOVA) (exercise × model) with Bonferroni post-hoc corrections was used to assess between-model differences and relevant interaction effects using absolute differences (direct 1RM—predicted 1RM) in addition to 95% confidence intervals (CI) and partial eta-squared ES ($\eta_{p\;}^{2}$). Where sphericity was violated (assessed via Mauchly’s tests of sphericity), the Greenhouse–Geisser correction was applied. Alpha level was set at *p* < 0.05. SPSS (24.0, IBM, New York, NY, USA) and Microsoft Excel was used for statistical analyses.

## 3. Results

All data were normally distributed and met the necessary assumptions prior to analysis, or appropriate corrections were applied. Measured 1RM was 157.0 ± 19.4 kg. Means, SDs, and 95% CIs of the predicted 1RM data can be found in Table 2. Practically perfect correlations (*r* > 0.97) were observed for all predictive models when compared to the measured 1RM data (Table 1). Back squat (quadratic 7) model yielded the largest SEE (4.06 kg), with the remaining models < 4 kg (Table 2). The four quadratic predictive models reported trivial ES (*g* = −0.06–0.04), compared to the linear models for the back squat and combined methods, which reported moderate (*g* = 0.52) and small ES (*g* = 0.12–0.40), respectively (Table 2).

The mean differences in model predicted and measured 1RM can be seen in Figure 2. The four quadratic models produced differences ranging from −1.2 to 0.7 kg, lower than that of the linear models, which ranged from 2.4 to 9.9 kg (Figure 2, Table 1). Small systematic biases were reported for all four quadratic models (−1.17–0.73 kg), with random error ranging from ± 3.09–7.67 kg, whereas the linear models all underestimated the predicted 1RM (2.37–9.87 kg), with random error of 5.11 to 6.34 kg being observed (Figure 3).

A significant two-way interaction was observed between exercise and model (*F*(1.65, 21.48) = 23.95, *p* < 0.001, $\eta_{p}^{2}\;$= 0.65), with simple main effects observed across models (combined: *F*_(2.01, 26.15)_ = 121.47, *p* < 0.001, $\eta_{p\;}^{2}\;$= 0.90; back squat: *F*_(1.93, 25.10)_ = 7.11, *p* = 0.004, $\eta_{p}^{2}$ = 0.35). When applying back squat only data, Bonferroni tests revealed significant differences between quadratic and linear models (4-point: 3.55 kg (95% CI: 0.22–6.88 kg), *p* = 0.034; 7-point: 2.93 kg (95% CI: 0.01–5.85 kg), *p* = 0.049), but no significant differences between 4-point and 7-point models (quadratic: 1.89 kg (95% CI: −1.55–5.34 kg), *p* = 0.670; linear: 1.27 kg (95% CI: −1.20–3.75 kg), *p* = 0.805). Post-hoc tests also revealed significant differences between quadratic and linear models (4-point: 8.52 kg (95% CI: 6.41–10.64 kg), *p* < 0.001; 7-point: 9.20 kg (95% CI: 7.23–11.17 kg), *p* < 0.001) and between the 4-point and 7-point linear models (2.14 kg (95% CI: 0.95–3.33 kg), *p* = 0.001), but not quadratic (1.46 kg (95% CI: −0.52–3.45 kg), *p* = 0.235) when utilizing the combined method.

Simple main effects were observed for exercise when applying linear modeling (7 point: *F*_(1, 13)_ = 51.56, *p* < 0.001, $\eta_{p}^{2}$ = 0.80; 4 point: *F*_(1, 13)_ = 26.60, *p* < 0.001, $\eta_{p}^{2}$ = 0.67), but not quadratic modeling (7 point: *F*_(1, 13)_ = 0.008, *p* = 0.929, $\eta_{p}^{2}$ = 0.001; 4 point: *F*_(1, 13)_ = 0.24, *p* = 0.632, $\eta_{p}^{2}$ = 0.18). Mean differences between exercises for linear models were 5.34 kg (95% CI: 3.11–7.58 kg) and 6.21 kg (95% CI: 4.34–8.08 kg) for 4-point and 7-point modeling, respectively, with quadratic models as 0.37 kg (95% CI: −1.27–2.01 kg) and 0.57 kg (95% CI: −1.29–1.41 kg) for 4-point and 7-point modeling, respectively.

## 4. Discussion

The aim of this study was to investigate whether 1RM could be predicted from load-velocity data. Specifically, to compare whether exercise selection (back squat vs. jump squat and back squat, ‘*combined*’ method) and model construction (linear vs. quadratic) effects the predictive validity of the LVP when using 80% 1RM as the model reference velocity. The main findings of this research were that 1RM could be accurately predicted from load-velocity data, and that quadratic modeling demonstrated a greater accuracy than linear modeling. Furthermore, when applying quadratic modeling to LVP data, the combined method was as accurate as the back squat condition, whereas significant differences were evident between the approaches with linear modeling.

The findings of this study (Table 2) support recent research highlighting the accuracy of using LVP data for maximum load estimation [17,18,24,25,28,36]. Despite this, our data did show discrepancies between linear modeled estimated 1RMs and measured 1RMs. Significant differences were observed for all four linear models, with mean differences ranging from 2.4–9.9 kg (Figure 2). When applied to free-weight, lower-body exercises, previous literature investigating the predictive validity of LVP data supports our findings. Ruf et al. [16], Lake et al. [15], and Banyard et al. [10] all reported inaccurate estimations of predicted 1RMs ranging from 5–40 kg (*p* < 0.05; ES = −1.24–1.04) in the deadlift and back squat. Interestingly, much smaller SEEs (2.2–3.3 kg vs. 10.6–17.2 kg) and systematic biases (2.4–9.9 kg vs. 20.0–30.9 kg) were observed in the present study compared to previous data [10]. These discrepancies may be partially explained by the differences in extrapolation methods applied. Earlier research utilized the V_1RM_ as the reference point for predictive modeling, despite research indicating its poor validity and reliability [11,16,22]. As a result, our models were based on the heaviest load (80% 1RM) that demonstrated acceptable levels of reliability (80% 1RM CV = 5.4–5.7% vs. V_1RM_ = 11.8–19.4%) [11,22]. Given the superior within-subject reliability of mean velocity associated with submaximal loads [11], it is likely that the magnitude of random error in our model was reduced.

Our predictive modeling involved a process of interpolation of a more reliable mean velocity (80% 1RM), followed by extrapolation from the estimated 80% 1RM to 1RM (in kgs), whereas previous literature has typically estimated 1RM via extrapolation up to the V_1RM_ [10,16,17,25,36]. The V_1RM_ method relies on the point of extrapolation aligning fully to the trend of the data, with the model required to capture the underlying values it estimates. Often, when that point of interest is the V_1RM_, the estimation can be compromised because the rate of change in velocity is not as constant (slope < 1) compared with sub-maximal loads. Instead, interpolation can account for this as the estimation of values fall inside the range of observed data, which is more likely to be captured by the model function, leading to less erroneous estimations. Finally, as relative (% 1RM) and absolute (kg) load are both ratio data, they scale proportionally, meaning our method of extrapolation from a predicted 80% to 100% 1RM seems more robust for maximum load estimation than extrapolation to V_1RM_. Future research should look to employ this method of estimation to other exercises to further investigate its predictive validity.

Previous literature applying linear modeling to LVP data has reported smaller differences and associated error than our study. Mean differences of <5 kg have been reported in the half squat and bench-press exercises from two-point and multiple-point methods [18,24,36], however, this research typically employs Smith-machine-based protocols. Despite numerous criticisms regarding Smith-machines and their transferability to applied settings, most literature in this space continues to employ them. Research suggests that mechanical outputs such as take-off velocity (directly related to peak velocity), maximum load lifted, and electromyographical muscle activity differ when performing Smith-machine exercises compared to free-weight, suggesting that the generalizability of this research to broader contexts using free-weight exercise is limited [37,38,39]. Future research should therefore seek to elucidate the predictive validity of approaches most represented in practice, such as free-weight upper and lower body exercises.

This is the first study to compare different LVP-based predictive modeling in a free-weight, lower body exercise. A significant two-way interaction was evident with significant main effects, with all linear models significantly underestimating 1RM in comparison to their quadratic counterparts (*p* < 0.05). Larger LOAs were also evident, irrespective of the exercise employed (Figure 3), indicating the superiority of quadratic modeling for estimating 1RM in the free-weight back squat. Interestingly, the only previous study testing similar hypotheses was in the Smith-machine bench press and reported multiple-point linear modeling as superior to second-order polynomial modeling [28]. A Smith-machine is designed to limit movement in the sagittal and frontal planes, potentially increasing the reliability of velocity data, and creating a more linear trend [38]. Similarly, lower-body movements are more complex in nature (more joints involved, greater displacement traveled, and a more varied bar path) than upper body (generally, a more vertical, linear bar path), requiring a greater interaction between joint angular forces, moments, and velocities, potentially resulting in a less predictable relationship [40]. Therefore, practitioners should use more sophisticated 1RM predictive models based on LVPs to account for the less-predictable nature of lower-body, free-weight exercises. In addition, no significant differences were observed in predictive validity based on the number of data points used to construct the profile in this study (2-point vs. multi-point) [28] as well as ours (4-point vs. 7-point), suggesting both models could be implemented effectively at the start of a training session to update daily 1RMs quickly with only a few loads lifted.

When applying linear modeling, a significantly larger mean difference and larger LOAs were observed for the combined vs. back squat method (*p* < 0.001). Conversely, no significant differences were observed between exercises when applying quadratic modeling, suggesting this model has a greater level of sophistication that can fit various types of LVP data. Previous research has reported greater mechanical output (velocity, force, power) when performing ballistic exercises using light-to-moderate loads compared to their non-ballistic counterparts, primarily because of the large period of negative work (braking) at the end of the concentric phase [31,33]. Despite this, LVPs are typically derived using non-ballistic exercise only, even when starting at 0–30% 1RM [11,22,25,41]. Capturing load-velocity data this way could be sub-optimal and less valid given the reported lower mechanical output [31,33]. Therefore, utilizing the combined method with quadratic modeling seems the most logical, valid, and effective way to construct a LVP and predict 1RM.

In contrast to previous literature, the current study assessed predictive validity by first constructing the model from initial testing data (i.e., collect LVP data and determine the 80% 1RM velocity), and then subsequently assessed its validity using newly collected data from a second session. This approach provides greater confidence that the predictive models can estimate future observations with suitable accuracy. Furthermore, the use of LVPs as a longitudinal tool relies on the stability of velocity at relevant percentages of 1RM, irrespective of physiological adaptations. Whilst scarce, previous literature suggests that mean velocity is stable following bouts of acute strength training (~4–6 weeks) [21,42,43], providing confidence in the predictive models. Future research, however, should seek to further investigate the stability of the LVP across longer time periods (e.g., full macrocycle) as well as predict 1RM over multiple sessions, as often, predictive models can be misleadingly concluded as valid and reliable when only applied to one session’s worth of data.

## 5. Conclusions

Prediction of 1RM based on LVP data might be an effective autoregulatory tool for S&C practitioners over the course of a training cycle. The results of this study provide practitioners with confidence that a quadratic model that uses mean velocity of 80% 1RM and utilizes both ballistic and non-ballistic exercises is an effective method for estimating an individual’s 1RM in the free-weight back squat, ensuring load manipulation and fatigue management can be achieved on a sessional basis. Given the nature of the protocol, it would also be feasible for a coach to employ this method at the beginning of a training session, estimate an athlete’s daily 1RM, adjust relevant working loads, and ensure parity between the loads prescribed and the intended training stimulus on that day. This would also allow coaches to utilize the integration of technology at the start of a training session, freeing up their time and attention for coaching for the remainder.

## Figures and Tables

**Figure 1 sports-09-00088-f001:**
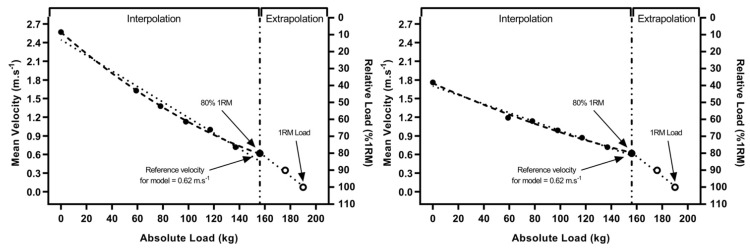
Visualization of the calculation method for the linear and quadratic one repetition maximum (1RM) predictive models for combined (left) and back squat (right) methods. Reference velocity was taken from session 1 and applied to session 2 data. Method of interpolation refers to the prediction of 80% 1RM absolute load (kg) from the LVP data model. Extrapolation refers to the prediction of 1RM absolute load (kg) from estimated absolute (kg) and relative (% 1RM) load data. Dotted lines indicate linear model, hashed line indicates quadratic model.

**Figure 2 sports-09-00088-f002:**
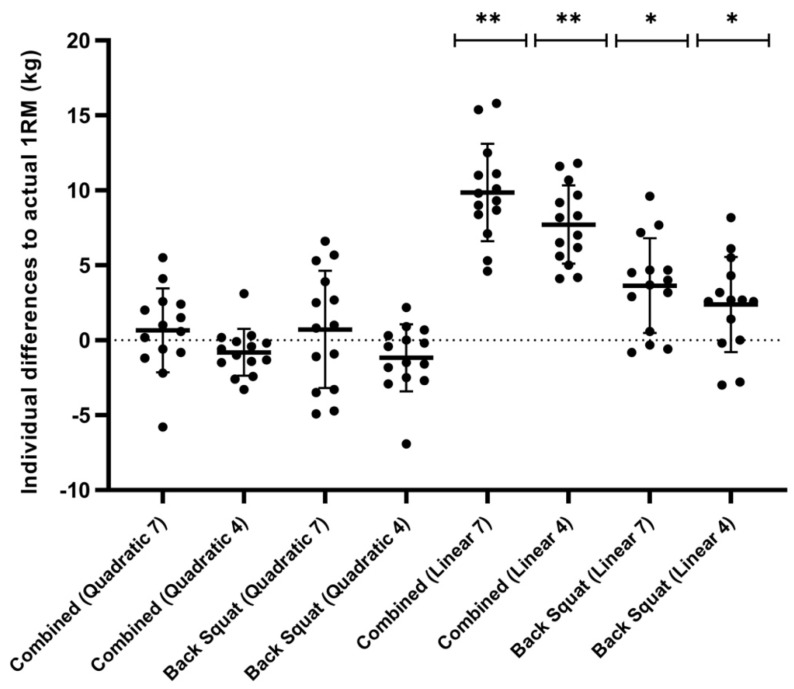
Individual and mean differences for one repetition maximum (1RM) predictive model vs. actual 1RM (represented as actual 1RM minus predicted 1RM). Horizontal lines indicate mean with SDs as error bars. Combined = jump squat and back squat method. 4 = 4 data-points; 7 = 7 data-points. ** (*p* < 0.001), * (*p* < 0.05).

**Figure 3 sports-09-00088-f003:**
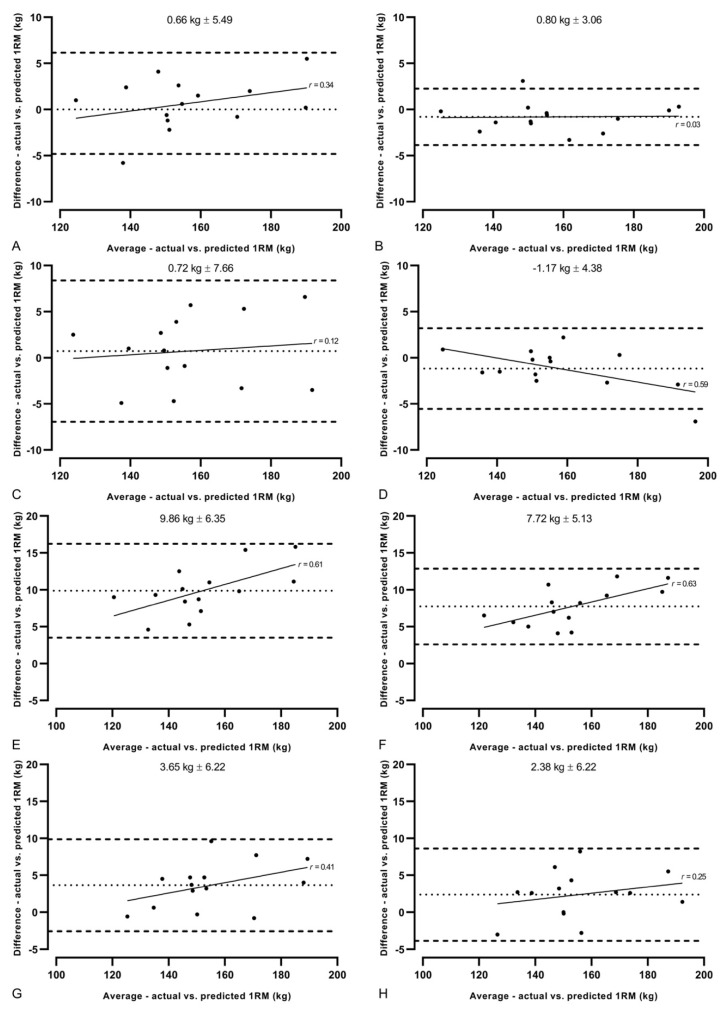
Bland–Altman plots for all eight one repetition maximum (1RM) predictive models. (**A**–**D**) quadratic models; (**E**–**H**) linear models; (**A**,**B**,**E**,**F**) combined method (jump squat and back squat); (**C**,**D**,**G**,**H**) back squat only method; (**A**,**C**,**E**,**G**) 7-point models; (**B**,**D**,**F**,**H**) 4-point models. Dotted lines indicate mean systematic bias. Hashed lines indicate 95% limits of agreement. Straight lines indicate heteroscedasticity of the models (linear regression) with *r* values labeled beside them. Values above figures represents mean systematic bias ±95% limits of agreement.

**Table 1 sports-09-00088-t001:** Description of all eight one repetition maximum (1RM) predictive models. All loads between and including 0% 1RM and 60% 1RM in the combined method were taken from jump squat data. Loads > 60% 1RM in the combined method were taken from back squat data.

Name	Model	Exercise	Data Points	Loads (% 1RM)
Combined (quadratic 7)	Quadratic	Jump Squat + Back Squat	7	0 load + 30–80%
Combined (quadratic 4)			4	0 load, 30%, 50%, 80%
Back Squat (quadratic 7)		Back Squat	7	0 load + 30–80%
Back Squat (quadratic 4)			4	0 load, 30%, 50%, 80%
Combined (linear 7)	Linear	Jump Squat + Back Squat	7	0 load + 30–80%
Combined (linear 4)			4	0 load, 30%, 50%, 80%
Back Squat (linear 7)		Back Squat	7	0 load + 30–80%
Back Squat (linear 4)			4	0 load, 30%, 50%, 80%

**Table 2 sports-09-00088-t002:** One repetition maximum (1RM) descriptive data (means and SD) with 95% confidence intervals (CI), Pearson correlation coefficient (*r*), standard error of the estimate (SEE), *p* values, and Hedges *g* effect sizes (+ 95% CI) for all eight predictive models. Measured 1RM = 157.0 ± 19.4 kg. 4 = 4-data points and 7 = 7-data points used to construct the model. *p* < 0.05.

Model-	Mean (kg)	SD (kg)	95% CI (kg)	*r*	SEE (kg)	*p*	Effect Size (*g*) + 95% CI
Combined (quadratic 7)	156.34	18.45	120.17–192.51	0.990	2.81	0.391	0.03 (−0.74, 0.81)
Combined (quadratic 4)	157.80	19.34	119.89–195.72	0.997	1.62	0.077	−0.04 (−0.82, 0.74)
Back Squat (quadratic 7)	156.27	18.94	119.15–193.40	0.979	4.06	0.502	0.04 (−0.74, 0.81)
Back Squat (quadratic 4)	158.17	20.70	117.60–198.75	0.996	1.82	0.071	−0.06 (−0.83, 0.72)
Combined (linear 7)	147.13	17.42	112.98–181.28	0.990	2.82	<0.001	0.52 (−0.27, 1.31)
Combined (linear 4)	149.27	17.72	114.53–184.01	0.994	2.19	<0.001	0.40 (−0.38, 1.19)
Back Squat (linear 7)	153.36	18.07	117.94–188.78	0.988	3.11	0.001	0.19 (−0.59, 0.97)
Back Squat (linear 4)	154.63	18.59	118.20–191.05	0.987	3.26	0.015	0.12 (−0.66, 0.90)

## Data Availability

Data is available upon request. Please contact the corresponding author.

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
