# Peer review of "A Novel Approach to 1RM Prediction Using the Load-Velocity Profile: A Comparison of Models"

_sports, 2021, doi:10.3390/sports9070088_

Round 1
Reviewer 1 Report
A very interesting study that investigates different 1RM prediction in squats.
It is not clear why 805 of 1RM is more accurate to use than 1RM velocity. This has to be explained more in detail. You would expect that lighter weights could be lifted with different mean velocities, while 1RM is limited to a minimum velocity otherwise you will not surpass the sticking region. This mean velocity is around 0.25m/s in squats. What would this then be for 80% of 1RM.
If I am correct shows table 2 the estimated 1RM calculated by the different methods. Then the linear models result in significantly lower 1RM numbers than the actual one, while further down it is written that all linear models overestimate the predicted 1RM. How is possible. Should not figure 2 show the opposite and also the whole discussion be changed in under estimation or is table 2 wrong?? This has to be changed in the whole manuscript to be sure that you discuss the right findings. The same for figure 3.
Specific comments:
Page 4 0%, 30% … Is it possible that there is 0% of 1RM. Rewrite. It is a jump squat without extra load, but it is not 0% of 1RM.
Page 5 … and standard error of the estimate (SEE)
Figure 3 put the a, B, etc above each figure.
Reviewer 2 Report
The study described in this manuscript compares models of predicting 1RM using the load-velocity profile.
Line numbers would be helpful.
The introduction section provides a comprehensive background on 1RM assessment and the importance of 1RM prediction from lighter lifts. However, i think more details as to why the authors think 1RM testing is not feasible for multifaceted sports. Is this a common problem for S&C coaches? Answering this question will help to support the need for your prediction model.
Under "procedures" please provide a sentence or two to explain the high-bar back squat technique.
Change "S&C coach..." to "Strength and Conditioning (S&C) coach..."
The mathematical modeling and statistical analyses are appropriate.
The Quadratic and Linear Model equations should be a figure rather than just floating between the written paragraph and Table 1.
The results are clearly explained. Good use of the tables and figures provided.
In the first line of page 10 change "compared to" to "compared with."
In the first line of "Conclusions" change "based off" to "based on."
It is also not good to start the sentence with the number 1. It would be better to start the sentence "Prediction of 1RM based on LVP data..."
